evolution, behaviour, theoretical biology

major transitions, evolution of eusociality, kin selection, parental manipulation, parent-offspring conflict, evolutionary dynamics

**Authors for correspondence:**
Mauricio González-Forero
e-mail: mgf3@st-andrews.ac.uk
Jorge Peña
e-mail: jorge.pena@iast.fr

†Both authors contributed equally.

# Eusociality through conflict dissolution

Mauricio González-Forero[1,†] and Jorge Peña[2,†]

[1]School of Biology, University of St Andrews, St Andrews, UK
[2]Institute for Advanced Study in Toulouse, University of Toulouse Capitole, Toulouse, France

(iD) MG-F, 0000-0003-1015-3089; JP, 0000-0002-4137-1823

Eusociality, where largely unreproductive offspring help their mothers reproduce, is a major form of social organization. An increasingly documented feature of eusociality is that mothers induce their offspring to help by means of hormones, pheromones or behavioural displays, with evidence often indicating that offspring help voluntarily. The co-occurrence of maternal influence and offspring voluntary help may be explained by what we call the converted helping hypothesis, whereby maternally manipulated helping subsequently becomes voluntary. Such hypothesis requires that parent-offspring conflict is eventually dissolved—for instance, if the benefit of helping increases sufficiently over evolutionary time. We show that help provided by maternally manipulated offspring can enable the mother to sufficiently increase her fertility to transform parent-offspring conflict into parent-offspring agreement. This conflict-dissolution mechanism requires that helpers alleviate maternal life-history trade-offs, and results in reproductive division of labour, high queen fertility and honest queen signalling suppressing worker reproduction—thus exceptionally recovering diverse features of eusociality. As such trade-off alleviation seemingly holds widely across eusocial taxa, this mechanism offers a potentially general explanation for the origin of eusociality, the prevalence of maternal influence, and the offspring's willingness to help. Overall, our results explain how a major evolutionary transition can happen from ancestral conflict.

## 1. Introduction

A few major evolutionary transitions in individuality have had vast effects on the history of life. Examples include transitions from prokaryotes to eukaryotes, from unicellularity to multicellularity, and from solitary life to eusociality. A major transition is said to occur when independently replicating units evolve into groups of entities that can only replicate as part of the group and that show a relative lack of within-group conflict [1–3]. A transition is envisaged to involve the formation of a cooperative group and its transformation into a cohesive collective [2,3]. These steps are hypothesized to occur through the evolution of cooperation, division of labour, communication, mutual dependence, and negligible within-group conflict, leading to a higher-level individual [3]. This scheme poses the question of how its various features can arise.

The transition to eusociality has been extensively studied, partly because it has occurred relatively recently. Eusociality is commonly defined as involving groups with reproductive division of labour, overlapping generations and cooperative work [4]. Additionally, an increasingly documented feature of eusociality is that mothers exert a substantial influence—via various proximate mechanisms—on whether offspring express helper phenotypes. Examples include hymenopteran queen pheromones suppressing worker reproduction [5], termite queen pheromones inhibiting differentiation of new queens [6], naked-mole rat workers becoming more responsive to pup calls after coprophagy of queen's faeces containing oestradiol [7] and queen presence suppressing gonadal development of females in eusocial shrimp [8]. This pattern suggests that explanations for the transition to eusociality should also account for the prevalence of maternal influence on helpers at the nest.

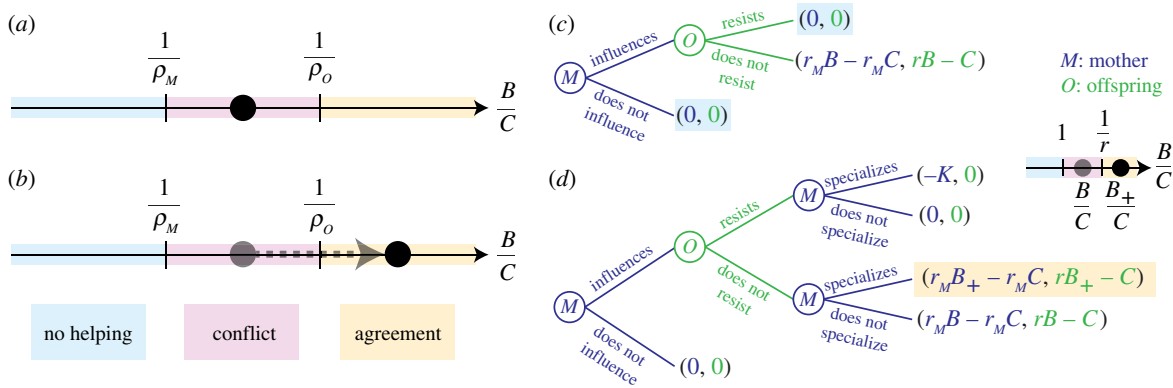

**Figure 1.** Conflict dissolution. (*a*,*b*) Helping is (i) disfavoured by mother and offspring if the benefit-cost ratio $B/C$ satisfies $B/C < 1/\rho_M$ (*no helping* zone); (ii) favoured by mother and offspring if $B/C > 1/\rho_O$ (*agreement* zone); or (iii) favoured by the mother but disfavoured by offspring if $1/\rho_M < B/C < 1/\rho_O$ (*conflict* zone). Conflict dissolution occurs when (*a*) $B/C$ starts in the conflict zone but (*b*) ends in the agreement zone. Helping is favoured by actors $A$ when $\rho_A B - C > 0$ (a Hamilton's rule; [9]), where $C$ is the cost to helpers, $B$ is the benefit to help recipients and $\rho_A$ is the *relative reproductive worth*, for actors $A$, of recipients relative to helpers (a reproductive-value weighted measure of relatedness; if all offspring are female, then $\rho_M = r_M/r_M = 1$ and $\rho_O = r/1 = r$, where $r_M$ and $r$ are the relatedness of a female to a daughter and a sister, respectively; see the electronic supplementary material, appendix, S3). (*c*,*d*) Sequential games modelling conflict and conflict dissolution via maternal reproductive specialization. (*c*) Without specialization, conflict yields equilibria with no helping (*shaded*); (*d*) with specialization, conflict is dissolved if $B_+/C > 1/\rho_O$, yielding a unique equilibrium under agreement (*shaded*). $K$ is the cost of specialization without helpers. (Online version in colour.)

Two classic hypotheses for the origin of eusociality offer different explanations for the prevalence of maternal influence. On the one hand, the *voluntary helping hypothesis* proposes that helping arises in the evolutionary interests of helpers, in the sense that helping is favoured when helpers have unconstrained control of their helping behaviour [9]. According to this hypothesis, helping evolves in simple models if $B/C > 1/r$, where $B$ is the benefit given by helping, $C$ is the cost paid for helping and $r$ is the relatedness of helper towards recipient. In this view, maternal influence on workers would arise as a regulatory mechanism after helping evolves, and the prevalence of such maternal influence would be a consequence of the loss of eusociality without it. On the other hand, the *maternal manipulation hypothesis* proposes that helping arises in the evolutionary interests of mothers against the evolutionary interests of helpers—that is, there is a parent-offspring conflict over helping [10–12]. In this case, helping evolves if $B/C > 1$, which is easier to satisfy than the condition for voluntary helping, as long as $r < 1$ [13]. Although, by definition, the maternal manipulation hypothesis would account for the prevalence of maternal influence, this hypothesis is refuted by increasing evidence suggesting that it is often in the evolutionary interests of offspring to help [14,15], thus supporting the voluntary helping hypothesis.

A third alternative hypothesis—that we term the *converted helping hypothesis*—proposes that helping initially arises from maternal manipulation but then becomes voluntary [16,17]. This hypothesis can bring together advantages of both the voluntary helping and maternal manipulation hypotheses without bringing in their disadvantages. First, because it is initially maternally manipulated, helping originates under the easier condition $B/C > 1$ and would be associated with maternal influence. Second, because converted helping is voluntary in the end, the hypothesis is also consistent with evidence that offspring help voluntarily. By considering that manipulated helping becomes voluntary, the converted helping hypothesis requires that there is a switch from conflict to agreement, that is, that *conflict dissolution* occurs

(figure 1*a*,*b*). Hence, it is of substantial interest to identify mechanisms that dissolve conflict and that would give the converted helping hypothesis a basis.

Here, we report a conflict-dissolution mechanism that yields eusociality together with its hallmarks of maternal influence on offspring helping phenotype, offspring voluntary helping and high maternal fertility. We term this particular mechanism *conflict dissolution via maternal reproductive specialization*, whereby (i) the mother manipulates offspring to become helpers (i.e. against their inclusive-fitness interests); (ii) while offspring evolve resistance to manipulation, the mother uses available help to become more fertile; and (iii) increased maternal fertility increases the benefit of helping to the point of rendering helping voluntary (i.e. in the inclusive fitness interest of helpers). The key requirement for this mechanism to work is that helpers alleviate the total per cent life-history trade-off limiting maternal fertility in the absence of help—a requirement that available evidence suggests may hold widely across eusocial taxa. We show how conflict dissolution via maternal reproductive specialization operates by means of both a heuristic game theory model and a demographically explicit evolutionary model.

## 2. Model and results

### (a) Sequential game

First, we use a sequential game to show that offspring resistance can prevent maternal manipulation from yielding helping. Consider a game between a mother ($M$) and a female offspring ($O$) (figure 1*c*). First, $M$ chooses between either influencing $O$ or not. Second, if $M$ influences $O$, then $O$ chooses between either resisting the influence or not. If $O$ does not resist, she helps $M$ produce an extra number $B$ of daughters, at a cost $C$ to herself. If $M$ is related to each daughter by $r_M$, and if $O$ is related to each sister by $r$, then $M$ gets an 'inclusive-fitness pay-off' of $r_M B - r_M C$ while $O$ gets $rB - C$. Otherwise, if $M$ does not influence or if $O$ resists, $O$ does

not pay any cost and no extra daughters are produced, yielding pay-offs of zero to both $M$ and $O$. Under conflict ($1 < B/C < 1/r$), maternal influence constitutes manipulation, selection favours resistance and manipulation does not yield helping: the game has two subgame perfect equilibria, one with resistance and the other without influence.

Let us extend this game to show that reproductive specialization allows maternal influence to yield helping despite possible resistance. Now, after $O$ moves, $M$ can choose between specializing into reproduction or not (figure 1d). If $O$ resists, $M$ pays a cost $K$ for exerting more reproductive effort owing to a life-history trade-off. If $O$ does not resist, $M$ produces an extra number of daughters $B_+$ at no cost provided that $O$ alleviates the trade-off faced by $M$. Importantly, if helping and specialization are synergistic enough that $B_+/C > 1/r$, then there is agreement with specialization although there is conflict without it. Thus, influence and specialization yield helping: the game has a unique subgame perfect equilibrium with influence, specialization and no resistance. This game suggests that if mothers can use offspring help to increase their fertility sufficiently, the underlying parent-offspring conflict can be dissolved.

## (b) Evolutionary model

We now formulate an evolutionary model to show that the evolution of maternal reproductive specialization can increase the benefit of helping to a point where conflict is dissolved. The model is age-, sex- and genotype-structured with explicit population and mutant-invasion dynamics [18,19], which allows us to derive rather than assume inclusive-fitness pay-offs (the model is fully described in the electronic supplementary material, appendix, S1). The genetic system is diploid or haplodiploid, and either both sexes or only females help; this covers the spectrum of known eusocial taxa (electronic supplementary material, appendix, figure S1; [20]). We consider a large population with overlapping generations, a fixed number of nesting sites, and a monogamous life cycle with two offspring broods, as follows. (i) Young parents produce $f_1$ first-brood offspring and with probability $s_M$ survive to old age to produce $f_2$ second-brood offspring. (ii) Each first-brood offspring of the helper sex becomes a helper with probability $p$ or disperses with probability $1 - p$; the number of helpers $h$ at the nest is hence proportional to $p$. All second-brood offspring disperse. (iii) Dispersing first-brood offspring (resp. second-brood offspring) survive dispersal with probability $s_1$ (resp. $s_2$). Surviving individuals mate singly at random and start a nest if nesting sites are available (electronic supplementary material, appendix, figure S2). We assume vital rates are such that (i) $f_2$ increases with maternal reproductive effort $z$ (e.g. number of ovarioles), (ii) there is a trade-off between survival and fertility, so that $s_M$ or $s_2$ decreases with $f_2$, and (iii) helpers increase mother or second-brood survival, so that $s_M$ or $s_2$ increases with $h$. A couple's expected number of reproductive first-brood (resp. second-brood) offspring is given by the couple's early productivity $\Pi_1 = (f_1 - h)s_1$ (resp. late productivity $\Pi_2 = s_M f_2 s_2$). We analyse the coevolutionary dynamics of offspring helping probability $p$ and maternal reproductive effort $z$. We let $p$ be under maternal, offspring, or shared control. Under shared control, $p$ is a joint phenotype [21] that increases with maternal influence $x$ (e.g. pheromone production) and decreases with offspring

resistance $y$ (e.g. receptor antagonist production). Reproductive effort $z$ is under maternal control. For simplicity, we assume that maternal influence and offspring resistance are costless. For the inclusive fitness interpretation of our results, we distinguish between different sets of individuals in a focal nest. In particular, we denote by $M$ the singleton whose only member is the mother, by $O_{a\ell}$ the set of sex-$\ell$ offspring produced in brood $a$ (with $a \in \{1, 2\}$, and $\ell \in \{♀, ♂\}$), and by $O_a$ the set of all $a$-th brood offspring (i.e. both male and female). Furthermore, we let $O \equiv O_1$ if both sexes help, and $O \equiv O_{1♀}$ if only females help.

## (c) Inclusive fitness effects

We find that, in agreement with inclusive fitness theory, each evolving trait $\zeta$ (where $\zeta \in \{x, y, z\}$ for shared control) is favoured by selection if and only if its inclusive fitness effect $\mathcal{H}_\zeta$ is positive (see the electronic supplementary material, appendix, S2 and S3). More specifically, the selection gradients quantifying directional selection acting on each trait are

$$S_x \propto \frac{\partial p}{\partial x}(\rho_M B - C), \tag{2.1a}$$

$$S_y \propto \frac{\partial p}{\partial y}(\rho_O B - C), \tag{2.1b}$$

$$S_z \propto \frac{\partial \Pi_2}{\partial f_2}, \tag{2.1c}$$

where the inclusive fitness effect of helping from the perspective of actors $A$ is $\mathcal{H}_p^A \propto \rho_A B - C$ with $A = M$ when helping is under maternal control, and $A = O$ when it is under offspring control. Here, $C = -\partial \Pi_1/\partial h = s_1$ is the marginal cost of helping, $B = \partial \Pi_2/\partial h$ is the marginal benefit of helping, and $\rho_A$ is what we term the *relative reproductive worth* for a random actor in set $A$ of a random candidate recipient of help in set $O_2$ relative to a random candidate helper in set $O$. Our measure of relative reproductive worth generalizes Hamilton's life-for-life relatedness [22] to allow for helpers and recipients of both sexes. It depends on the relatedness of actors towards candidate recipients of help, the sex-specific reproductive values of such recipients, and the stable sex distribution of the parents of candidate helpers (electronic supplementary material, appendix, S3).

## (d) Conflict dissolution

We model the evolutionary dynamics after the canonical equation of adaptive dynamics [23–25] with selection gradients given by equation (2.1). Numerical solutions of the evolutionary model show that conflict dissolution via maternal reproductive specialization can occur. If maternal influence $x$ and offspring resistance $y$ coevolve under conflict but reproductive effort $z$ cannot evolve (i.e. there is no genetic variation for $z$), resistance may win the ensuing arms race and eliminate helping in the long run (figure 2a–e). This matches the standard expectation when maternal influence is carried out with pheromones [26–28]. Alternatively, if reproductive effort coevolves with influence and resistance, the benefit-cost ratio can move out of conflict and into the agreement zone (figure 2f–j). In this case, the arms race vanishes as manipulated helping becomes voluntary. The final outcome is eusociality where (i) helpers are maternally induced to help and not favoured to resist, and (ii) the mother has become highly fertile and reliant on helpers for her own or her offspring's survival. Moreover, ancestral manipulation

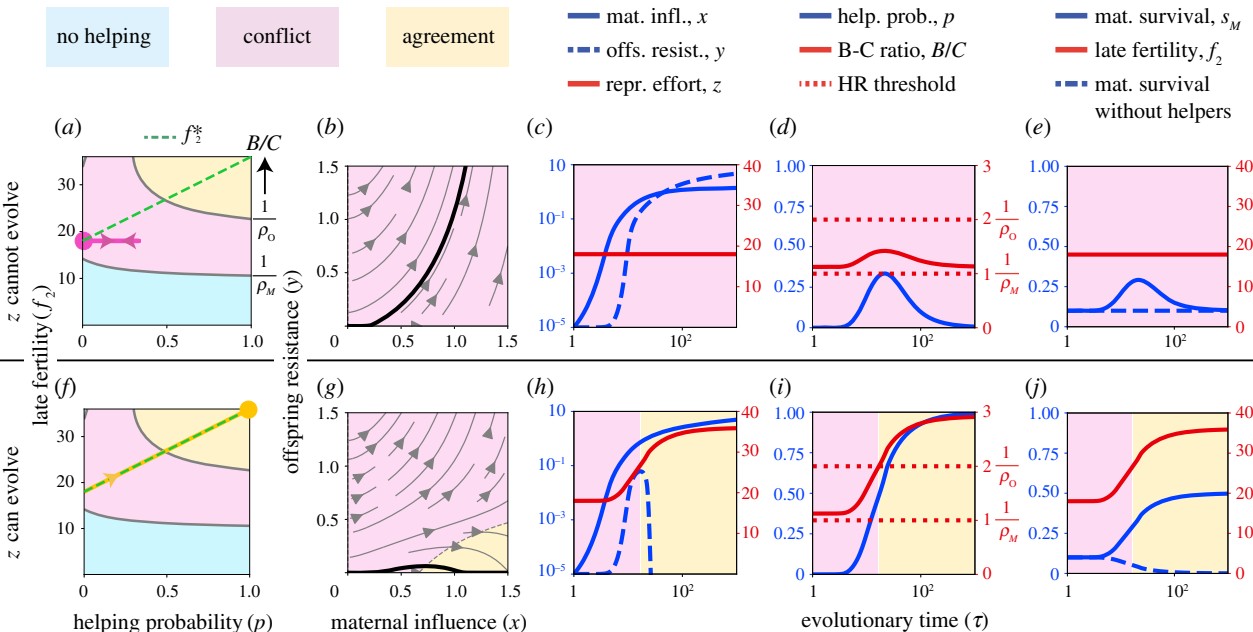

**Figure 2.** Conflict dissolution via maternal reproductive specialization (evolutionary model). (*a*–*e*) Coevolution of maternal influence $x$ and offspring resistance $y$ when maternal reproductive effort $z$—and hence late fertility $f_2$—cannot evolve (i.e. the genetic variance of $z$, $G_z$, is zero). (*a*) Phase portrait showing the evolution of the helping probability $p$ under constant late fertility $f_2$. Starting from conflict, helping evolves temporarily but is eventually lost owing to the evolution of resistance (start and end points are given by the *circle*; the *pink* trajectory ends in the conflict zone). (*b*) Stream plot showing the coevolution of maternal influence and offspring resistance. The *thick* line shows the trajectory for the initial conditions used. (*c*–*e*) Time series of: (*c*) the evolving traits; (*d*) the resulting helping probability $p$, benefit-cost ratio $B/C$, and the Hamilton's rule (HR) threshold from the mother and offspring perspective; and (*e*) the vital rates $s_M$, $f_2$ and $s_M$ with zero helpers. (*f*–*j*) Analogous plots but now $z$ can evolve as the mother chooses it optimally for the number of helpers she has (i.e. as if $G_z \to \infty$). In this case, fertility evolves along the optimal path, $f_2^*$. (*f*) Phase portrait showing the coevolution of the helping probability $p$ and optimal late fertility $f_2^*$. Starting from conflict, helping emerges and is maintained through the evolution of $z$ yielding agreement (end point is given by the *circle*; the *yellow* trajectory ends in the agreement zone). (*g*) Stream plot showing the coevolution of maternal influence and offspring resistance. The thick line shows the trajectory for the initial conditions used; such a trajectory starts at conflict but converges to agreement. (*h*) Resistance reversal. (*i*) $B/C$ evolves and the Hamilton's rule threshold from the offspring perspective is crossed. (*j*) The mother becomes highly fertile and reliant on helpers for her own survival. The genetic system is diploid, both sexes help, and helping is under shared control with sequential determination of the joint helping phenotype. Here, the life-history trade-off is between maternal survival $s_M$ and late fertility $f_2$, as illustrated in figure 3. Second-brood offspring survival $s_2$ is constant. The remaining details of the functional forms and parameter values used are given in the electronic supplementary material, appendix, S8. (Online version in colour.)

becomes an honest signal [29]: the resulting maternal influence alters the recipient's phenotype in the recipient's interest (i.e. helpers are induced to help, and they 'want' to help); the signaller evolved to produce that effect (i.e. maternal influence evolved to induce helping); and the recipient evolved to attend the signal (i.e. offspring evolved lack of resistance to influence).

## (e) Trade-off alleviation

We now show that conflict dissolution via maternal reproductive specialization requires that helpers alleviate the total per cent trade-off limiting maternal fertility. Conflict occurs when the mother favours helping (i.e. $\mathcal{H}_p^M > 0$) while offspring disfavour helping (i.e. $\mathcal{H}_p^O < 0$). Conflict dissolves if there is eventual agreement (i.e. $\mathcal{H}_p^M > 0$ and $\mathcal{H}_p^O > 0$ in the end). Hence, for conflict dissolution to occur it is necessary that the inclusive fitness effect $\mathcal{H}_p^O$ for helping under offspring control increases with evolutionary time $\tau$ and changes sign from negative to positive, namely that

$$\frac{\mathrm{d}\mathcal{H}_p^O}{\mathrm{d}\tau} > 0 \text{ for all } \tau \in [\tau_1, \tau_2] \text{ and} \quad \text{(persuasion condition)}$$

$$\mathcal{H}_p^O = 0 \text{ for some } \tau \in (\tau_1, \tau_2) \quad \text{(conversion condition)}$$

hold for some evolutionary time interval $[\tau_1, \tau_2]$. By the chain rule, the persuasion condition is equivalent to $(\partial \mathcal{H}_p^O / \partial p)$

$(\mathrm{d}p/\mathrm{d}\tau) + (\partial\mathcal{H}_p^O/\partial z)(\mathrm{d}z/\mathrm{d}\tau) > 0$ for all $\tau \in [\tau_1, \tau_2]$. Motivated by this, we say that conflict dissolution via maternal reproductive specialization occurs when $(\partial\mathcal{H}_p^O/\partial z)(\mathrm{d}z/\mathrm{d}\tau) > 0$ for all $\tau \in [\tau_1, \tau_2]$. Thus, conflict dissolution via maternal reproductive specialization requires that there is helping-fertility synergy (i.e. $\partial\mathcal{H}_p^O/\partial z > 0$; [30]) as reproductive effort increases over evolutionary time.

Helping-fertility synergy at an optimal fertility $f_2^*$ (implicitly given by $\partial\Pi_2/\partial f_2|_{f_2=f_2^*} = 0$) is equivalent to the four following statements (electronic supplementary material, appendix, S5). First, the benefit-cost ratio, $B/C$, increases with late fertility at an optimal late fertility $f_2^*$, so $\partial(B/C)/\partial f_2|_{f_2=f_2^*} > 0$. Second, optimal late fertility $f_2^*$ increases with the number of helpers, so $\mathrm{d}f_2^*/\mathrm{d}h > 0$. Third, the late productivity function $\Pi_2$ is supermodular, meaning that helping and fertility act as strategic complements, so that $(\partial^2\Pi_2/\partial f_2\partial h)_{f_2=f_2^*} > 0$ holds. Fourth, helpers alleviate the total per cent trade-off at optimal late fertility, so that

$$\left(\frac{\partial}{\partial h}[\epsilon_{f_2}(s_M) + \epsilon_{f_2}(s_2)]\right)_{f_2=f_2^*} > 0 \quad \text{(alleviation condition)}$$

holds, where

$$\epsilon_X(Y) = \frac{X}{Y}\frac{\partial Y}{\partial X} = \frac{\partial \ln Y}{\partial \ln X} \quad (2.2)$$

*Proc. R. Soc. B* **288**: 20210386

is the elasticity of $Y$ with respect to $X$ (i.e. the per cent change in $Y$ caused by a marginal per cent increase in $X$ [31]). The elasticities $\epsilon_{f_2}(s_M)$ and $\epsilon_{f_2}(s_2)$ measure the assumed per cent life-history trade-offs (i.e. that a normalized increase in late fertility causes a normalized decrease in either maternal or offspring survival) and consequently satisfy $\epsilon_{f_2}(s_M) < 0$ or $\epsilon_{f_2}(s_2) < 0$. The quantity $\epsilon_{f_2}(s_M) + \epsilon_{f_2}(s_2) < 0$ thus measures the total per cent life-history trade-off, with the alleviation condition stating that such trade-off must be less negative with marginally more helpers (figure 3). We conclude that a key requirement for conflict dissolution via maternal reproductive specialization is that the total per cent life-history trade-off faced by mothers with an optimal fertility is less severe with marginally more helpers.

## (f) Promoters of conflict dissolution

Conflict dissolution depends on the relative evolutionary speeds of the coevolving traits, as speeds determine the size of the basin of attraction towards agreement [16]. Conflict dissolution is thus promoted by higher genetic variance of maternally controlled traits and lower genetic variance of offspring-controlled traits (figure 4a,b). The power of mother and offspring on determining the joint phenotype [32] also affects the evolutionary speed (but not the direction of selection) of influence and resistance. Hence, conflict dissolution is promoted by high maternal power (figure 4c). Finally, the evolutionary speed depends on whether mother and offspring contest the joint phenotype simultaneously (e.g. behaviourally, through aggression [33,34]) or sequentially (e.g. physiologically, where the mother alters offspring development through nutrition or hormones transferred before eclosion or birth [35,36]). Conflict dissolution is promoted by simultaneous contests if resistance is small (figure 4d; see the electronic supplementary material, appendix, S7).

## 3. Discussion

We have shown that maternal reproductive specialization can dissolve conflict and yield a major transition. Conflict dissolution occurs here because of the evolutionary synergy between offspring help and maternal fertility, whereby the benefit of helping increases to a point that the original parent-offspring conflict shifts to parent-offspring agreement. This provides a widely relevant mechanism for the converted helping hypothesis to explain the origin of eusociality and various hallmarks thereof. As we now discuss, this hypothesis, where ancestrally manipulated helping eventually becomes voluntary, brings together advantages of both the voluntary helping [9] and maternal manipulation [10,11] hypotheses without bringing in their disadvantages.

The converted helping hypothesis brings advantages in that eusociality arises under less stringent conditions than under voluntary helping, while being supported by the available evidence supporting both voluntary helping and maternal manipulation. First, by being initially manipulated, converted helping requires smaller benefit-cost ratios than voluntary helping at the start of the evolutionary process. Second, converted helping co-occurs with maternal influence. Thus, the converted helping hypothesis is consistent with the widespread maternal influence observed across eusocial taxa. By contrast, widespread maternal influence is not necessarily expected from ancestral voluntary helping. Third, by being

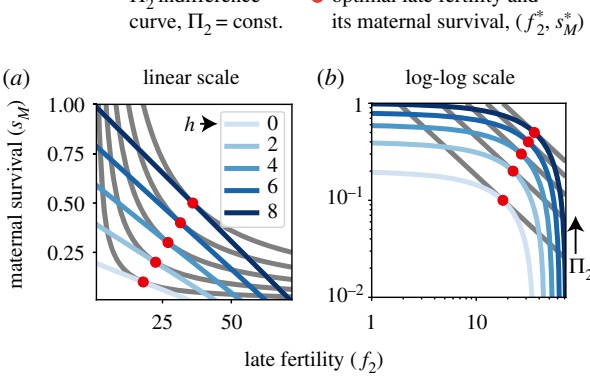

**Figure 3.** Survival-fertility trade-off alleviation by helpers. Here, maternal survival $s_M$ decreases with late fertility $f_2$ owing to the assumed trade-off (blue lines; linear trade-off in (a) linear scale or (b) log-log scale) whereas second-brood survival $s_2$ is constant. For a given number of helpers $h$, an optimal late fertility $f_2^*$ (red dots) occurs when a $s_M$ curve has the same slope as a $\Pi_2$ indifference curve (grey lines, where late productivity $\Pi_2$ is constant). In log-log-scale, all $\Pi_2$ indifference curves have the same slope, namely $-1$, as $\partial\Pi_2/\partial f_2 = 0$ is equivalent to $\epsilon_{f_2}(s_M) + \epsilon_{f_2}(s_2) = -1$ (see the electronic supplementary material, appendix, S5). Since here, $\epsilon_{f_2}(s_2) = 0$, the alleviation condition states that (in log-log scale) the slope of $s_M$ with respect to $f_2$ increases with increasing $h$ at an optimal $f_2^*$. Equivalently, the alleviation condition states that $f_2^*$ increases with $h$ (red dots move to the right as $h$ increases). Functional forms and parameter values are as in figure 2. (Online version in colour.)

eventually voluntary, converted helping requires high relatedness of helpers towards help recipients. Hence, the converted helping hypothesis is consistent with evidence that eusociality originated exclusively under lifetime monogamy [14].

In turn, the converted helping hypothesis does not bring disadvantages in that it is not refuted by the available evidence of voluntary helping refuting the maternal manipulation hypothesis. First, by turning manipulated helping into voluntary helping, conflict dissolution eliminates selection for resistance that would prevent the evolution of eusociality [26]. Second, because conflict dissolution turns manipulation into honest signalling, the converted helping hypothesis is consistent with evidence in extant taxa that queen pheromones act as honest signals rather than as manipulative control [5,15,26,28].

Although converted helping initially requires smaller benefit-cost ratios than voluntary helping, conflict dissolution is not necessarily straightforward. Indeed, conflict dissolution has additional conditions other than Hamilton's rule (e.g. the persuasion condition and conversion condition) and occurs under restricted parameter combinations (e.g. figure 4). This is in principle consistent with the patchy taxonomic distribution of eusociality, including the absence of eusociality in vast numbers of species with high intra-colony relatedness [37].

We distinguish conflict *dissolution*, which is the switch from conflict to agreement, from conflict *resolution*, which is the outcome of conflict even if conflict persists [38]. Conflict resolution is a static concept where it is enough to study evolutionary equilibria (e.g. evolutionarily stable strategies), whereas conflict dissolution is an out-of-equilibrium concept that requires an explicit consideration of the evolutionary dynamics. Thus, to establish that conflict dissolution has occurred, it is not sufficient to know that a population is at an agreement equilibrium, as the population may or may

**Figure 4.** Promoters of conflict dissolution. Resistance wins (trajectory ending at the *pink circle*) or conflict dissolution occurs (trajectory ending at *yellow circle*), respectively for (*a*) low or high genetic variance of reproductive effort, (*b*) low or high genetic variance of maternal influence, (*c*) low or high maternal power, and (*d*) sequential or simultaneous determination of the joint helping phenotype. The genetic system is diploid and both sexes help. Functional forms and parameter values are as in figure 2 except as follows. For (*a*), $G_z = 225$ for low genetic variance of $z$ and $G_z = 250$ for high genetic variance of $z$. For (*b*), $G_x = 0.9$ for low genetic variance of $x$ and $G_x = 1$ for high genetic variance of $x$ (and $G_z = 250$ for both). For (*c*), $\chi = 0.9$ for low maternal power and $\chi = 1$ for high maternal power (and $G_z = 250$ for both). For (*d*), sequential contest and simultaneous contest (and $G_z = 225$ for both). A very high genetic variance of $z$ is used here for visualization, but is not necessary for conflict dissolution (cf. electronic supplementary material, appendix, figure S14). (Online version in colour.)

not have arrived to the equilibrium from the conflict zone. Instead, one must consider initial conditions and the basins of attraction to agreement. For instance, worker reproduction in *Melipona* bees has been found to match the predicted optimum from the worker's perspective rather than the queen's perspective (fig. 2 of [39] and fig. 4 of [28]). Such match between conflict resolution models and empirical data suggests that helping is voluntary at present, but it is insufficient to rule out that helping was originally manipulated and only later became voluntary. In this sense, conflict dissolution depends on the evolutionary history, whereas conflict resolution is independent of it.

A key requirement of conflict dissolution via maternal reproductive specialization is that helpers alleviate the total per cent life-history trade-off limiting maternal fertility (i.e. the alleviation condition). This may hold widely as suggested by available empirical evidence. Indeed, data from eusocial bees, wasps and ants [40–42], as well as from cooperatively breeding mammals [43,44] and birds [45], indicate that the fertility of the breeding female often increases with the number of helpers. If such fertility is approximately optimal given the number of helpers available, these common empirical observations indicate that the alleviation condition may hold widely across eusocial taxa.

In another front, empirical inference of conflict dissolution may use its dependence on evolutionary history. In particular, conflict relics may be indicative of conflict dissolution [17]. For instance, the complex chemical composition of honeybee queen mandibular pheromone (QMP; which inhibits worker reproduction) suggests that it resulted from an arms race [46] that seemingly halted because (i) worker reproduction follows the workers' inclusive fitness interests [28,39], (ii) QMP behaves as an honest signal [15,47], and (iii) QMP composition is similar among related species [28,48]. By stemming from a halted arms race, QMP may be a conflict relic suggesting that conflict dissolution occurred.

Our mathematical model is related to previous models showing how the coevolutionary dynamics of multiple traits can make manipulated helping become voluntary

[16,17] (see also [49–51] for similar ideas in other systems). These models show that maternal manipulation can trigger not only the evolution of helper resistance but also the evolution of helper efficiency [16] or of the reduction of maternal care [17]. The evolution of these traits can make the benefit-cost ratio increase sufficiently over evolutionary time for voluntary helping to become favoured. In a similar vein, we have shown that manipulation can trigger the evolution of maternal reproductive specialization, which can make the benefit increase sufficiently for conflict to shift to agreement. While our mechanism requires the alleviation condition, which empirical evidence suggests may hold widely [40–45], available empirical evidence remains seemingly less supportive of other previously reported conflict-dissolution mechanisms [16,17]. Specifically, those mechanisms did not yield high maternal fertility and had more restrictive requirements, namely costly helping inefficiency [16] or better help use by maternally neglected offspring [17].

Eusociality through conflict dissolution via maternal reproductive specialization contains all the ingredients of a major transition [3]. First, cooperation evolves, specifically under relatively lax conditions because it is triggered by maternal manipulation. Second, division of labour evolves as the mother specializes in reproduction while offspring help in tasks such as colony defence, brood care and foraging. Third, honest communication evolves owing to conflict dissolution as manipulation becomes honest signalling. Fourth, mutual dependence evolves as the queen becomes unable to survive or reproduce without helpers (figure 2*j*). Fifth, negligible within-group conflict evolves because dissolution eliminates the parent-offspring conflict. Yet, our model did not let adults reproduce asexually in their natal nest. Such a conflict might persist in haplodiploids but can be removed by subsequent evolution of multiple mating and worker policing (as reviewed in [3]).

Conflict dissolution theory suggests that manipulation might play a role in explaining the empirically observed relevance of how groups are formed. Major transitions are envisaged to involve two steps, namely group formation

and group transformation [2,3]. How group formation occurs is thought to be key for major transitions to ensue, because both obligate multicellularity and eusociality have occurred by the staying together, and not the coming together, of lower-level entities [3]. Group formation matters in that staying together typically leads to higher relatedness relative to coming together, yet coming together can lead to high relatedness [52] but has seemingly not led to a major transition. This suggests that high relatedness alone is insufficient to explain why group formation is crucial. A contributing factor may be that staying together provides a stage for manipulation: staying together creates a power asymmetry, possibly giving the maternal entity an advantage at the very least by being there first. Even in clonal groups which lack genetic conflict between group members, such power asymmetry may be exploited by parasitic genetic elements seeking to promote their own transmission (owing to different transmission patterns among transposons, nuclear genes and cytoplasmic genes, or owing to different relatedness coefficients [53]). A parasitic genetic element might gain control of the division machinery of its host cell, keep daughter cells together and exploit them for its own benefit. This might occur against the interests of the host cell (i.e. with $B < C$ from the cell's perspective), possibly releasing an arms race [54]. However, in analogy to our results, such manipulation might also release the evolution of some form of specialization, eventually dissolving conflict between host and parasite, yielding a mutualism.

Although group formation and transformation are seen as occurring sequentially [3], our results indicate that they may reinforce each other. Group formation is seen as occurring first, whereby conflict is reduced [3]. Subsequently, group transformation, involving the evolution of division of labour, is seen as following [3]. By contrast, our model shows that after some incipient group formation via manipulation, group transformation can ensue via maternal reproductive specialization, which can then feed back to increase selection for helping. This positive feedback between helping and division of labour triggered by manipulation can dissolve conflict and generate a major transition from solitary living to eusociality.

Our results suggest how other major transitions might occur via similar mechanisms. Both the possibility of manipulation and the alleviation by manipulated parties of trade-offs faced by manipulating parties can occur in multiple settings. Additionally, subsequent interest alignment may occur not only through kin-selected benefits, but also through direct benefits. Thus, conflict dissolution may not only apply to fraternal but also to egalitarian major transitions [55]. Furthermore, conflict dissolution is likely to be important in cultural evolution. For instance, tax in its earliest forms constituted enforced labour [56], although tax compliance is now voluntary to a large extent in developed economies [57]. Voluntary tax compliance might stem from initial exploitation by monopolist rulers, triggering cultural evolution (e.g. of societal benefits) that dissolved conflict to some extent (e.g. as personal ethics evolve leading many subjects to eventually want to pay tax).

To conclude, our results offer a widely relevant mechanism for a unified hypothesis for the origin of eusociality and diverse features thereof, and suggest a reinterpretation of available evidence. More generally, analogous mechanisms of conflict dissolution operating during evolutionary, cultural or behavioural timescales may help understand how agreement can arise from conflict in other contexts.

**Data accessibility.** The code used for creating the figures of this paper is publicly available on GitHub (https://github.com/jorgeapenas/conflictdissolution).

**Authors' contributions.** M.G.F. and J.P. conceived the study, designed the model, derived the results and wrote the paper.

**Competing interests.** We declare we have no competing interests.

**Funding.** M.G.F. acknowledges funding from St Andrews' School of Biology. J.P. acknowledges IAST funding from the French National Research Agency (ANR) under the Investments for the Future (Investissements d'Avenir) programme, grant no. ANR-17-EURE-0010.

**Acknowledgements.** We thank I. Alger, A. Gardner, G. Nöldeke, P. Rautiala and two anonymous reviewers for feedback, E.J. Duncan and S. Giaimo for discussion, and A. Elbakian for the literature access.

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
