## [Peer Review File · Proceedings of the Royal Society B: Biological Sciences]

Review History

RSPB-2020-2980.R0 (Original submission)

Review form: Reviewer 1

Recommendation

Accept with minor revision (please list in comments)

Scientific importance: Is the manuscript an original and important contribution to its field?

Excellent

General interest: Is the paper of sufficient general interest?

Good

Quality of the paper: Is the overall quality of the paper suitable?

Excellent

Is the length of the paper justified?

Yes

Should the paper be seen by a specialist statistical reviewer?

No

Do you have any concerns about statistical analyses in this paper? If so, please specify them explicitly in your report.

No

It is a condition of publication that authors make their supporting data, code and materials available - either as supplementary material or hosted in an external repository. Please rate, if applicable, the supporting data on the following criteria.

Is it accessible?

N/A

Is it clear?

N/A

Is it adequate?

N/A

Do you have any ethical concerns with this paper?

No

Comments to the Author

This paper proposes and models a pathway for the evolution of eusociality. Most research in this area has focused on kin selection, with a minority emphasizing parental manipulation. Those are often treated as mutually exclusive explanation, but it also seems possible that some combination of them is required. This paper explores one such way of combining them. Here, the first step is parental manipulation to get offspring to stay and help against their kin-selected interests. In the next step, there are two options. In one, the offspring evolve to resist the manipulation and the incipient eusociality disappears. In the other, the mother (or mother-father pair) evolve towards optimizing her life history to the new situation of having some helpers. This, in some situations, alters the kin-selected B/C ratio of the offspring such that it now pays for them to stay and help. Eusociality is then selectively favored with no conflict, though the parental manipulation may persist as an honest signal.

The advantages of this model are nicely summarized by the authors. The initial threshold for manipulated helping is easier to achieve than the offspring kin selected one. Offspring acquiescence (under the altered B/C ratio) is most likely to be favored with high relatedness, consistent with the finding that single mating is the rule at the origin of eusocial taxa. The transition to honest signals from parental manipulation mechanisms is consistent with the kinds of signals observed in current social insects (although of course they could alternatively be honest signals without a manipulation precursor). And finally, a great advantage over the straight manipulation model is that the incentive for offspring to reverse the manipulation disappears and everyone agrees that helping is good. Overall, this model provides a potentially significant advance on a long-standing important question.

I think related ideas are probably out there in the literature, but maybe the idea has just been in my own head, as I'm not sure what citations to reference. One such idea is that maternal manipulation may immediately shift offspring choices such that helping is now favored. For example, by feeding an offspring less, the offspring may now be more likely to fail as a reproductive and, provided she is not equally more likely to fail as a helper (Craig 1979, 1983), that can make it pay to help instead. Craig was addressing West-Eberhard's "subfertility" hypothesis and I don't remember if they were explicitly invoking manipulation or not, but those papers might lead to others that do. Nevertheless, I see the current paper as a bit different. Instead of manipulation kicking offspring directly into the helping zone, it requires some further evolution of life history in the model.

I do not have any major reservations about the paper. Here are some minor comments:

Abstract (and elsewhere). The meaning of "requires that helpers alleviate the total percent life-history trade-off limiting maternal fertility" is not very clear to me. Can this be

reworded in some way?

Figure 1. Good figure in general, though I don't think K is defined.

consistent with single-mating origins in a way that total parental manipulation is not

I have some issues with the treatment of other models in the Discussion. However, none of these issues detracts from the main conclusions of this paper.

You argue that rigorous derivation is needed because inclusive fitness theory can mislead (line 259 ff). That is certainly true; using inclusive fitness thinking can be tricky. But I am not convinced by your example of the monogamy hypothesis. I think the differing conclusions of your model relative to the monogamy hypothesis represent differences in assumptions. In your two-generation cycle, the decision to reproduce versus help in generation 1 means that either you or some additional siblings will be reproductives in generation 2. Therefore the relevant r 's are 1 and $\frac{1}{2}$. But I think proponents of the monogamy hypothesis are envisioning a different life history. Suppose you emerge in your mother's nest and your choice is 1) leave and begin reproducing now or 2) stay and begin rearing your siblings now. The relevant exchange is offspring for siblings and the relevant r 's are $\frac{1}{2}$ and $\frac{1}{2}$. In other words, in your model, the option of reproducing directly involves a delay; you wait for the next generation (instead of rearing siblings in this generation). The upshot of that, with respect to relatedness, is that your relatedness through sibling help gets diluted by $\frac{1}{2}$ by the extra generation.

So I think the value of rigorous models versus inclusive fitness is more nuanced. Yes, in some sense the last word rests with the rigorous models, if they get their assumptions right and do not falsely exclude reasonable alternatives. But those models can be so complex (most kin selection researchers will not be able to follow your models in detail) that we run the risk of taking their results as more definitive than they are. Inclusive fitness reasoning, because it is so simple (though, yes, sometimes tricky), can provide a check on the rigorous models as well as a way for non-mathematicians to understand their results.

Parenthetically, as you will probably realize, your conflict dissolution ideas should not be ruined by the life history assumed by the monogamy model. There is still conflict because the mother's r 's have to be adjusted too (r to reproducing offspring $\frac{1}{2}$; r to that offspring's offspring $\frac{1}{4}$).

With perhaps somewhat less confidence, I would suggest that similar arguments might apply to your conclusions about the Taylor-Frank method (line 273 ff). This is suggested by your statement that it is unable to identify the logic problem with the monogamy hypothesis, which I have argued is not a logic problem at all. Again, I suspect that is true that Taylor-Frank modelers "subjective" choices about fitness are not necessarily subjective but may reflect assumptions of their model, though perhaps not always explicitly stated. Again, I would agree that it is great to have more rigorous models, but that they too can be misleading. Also, you are right that the T-F approach takes a direct-fitness perspective, but doesn't yours also take such a perspective from the beginning? Don't all "rigorous" models start that way and then have to reason themselves into a shifted inclusive perspective. That is commonly done with the T-F method as well.

Review form: Reviewer 2

Recommendation

Major revision is needed (please make suggestions in comments)

Scientific importance: Is the manuscript an original and important contribution to its field?

Acceptable

General interest: Is the paper of sufficient general interest?

Good

Quality of the paper: Is the overall quality of the paper suitable?

Marginal

Is the length of the paper justified?

Yes

Should the paper be seen by a specialist statistical reviewer?

No

Do you have any concerns about statistical analyses in this paper? If so, please specify them explicitly in your report.

No

It is a condition of publication that authors make their supporting data, code and materials available - either as supplementary material or hosted in an external repository. Please rate, if applicable, the supporting data on the following criteria.

Is it accessible?

Yes

Is it clear?

Yes

Is it adequate?

Yes

Do you have any ethical concerns with this paper?

No

Comments to the Author

Dear Editor, Proceedings B,

The paper entitled "Eusociality through conflict dissolution" by Gonzalez-Foreo and Pena is a theoretical paper investigating the origin of eusociality (sterile individuals helping the reproductives produce offspring under the division of labor, with cooperative interactions and overlapping generations). One of the hypotheses for the evolution of eusociality posits that in an evolutionary timescale, non-reproductive individuals (hereafter referred to as "helpers") originally resulted from the reproductives ("queens") manipulating them but at some point switched to work voluntarily: the authors mentioned this as "the converted helping hypothesis." One of the prerequisites of this hypothesis is queen-helper (or more specifically parent-offspring) has reached an agreement over evolutionary conflicts. Because of relatedness asymmetry, the direction of selection for the voluntary working may differ for the two (or more) classes of individuals, but if the direction of selection is (always) the same between the classes, the conflict of interests would no longer exist. Their question is how this agreement can reach and if this mechanism may be the origin of eusociality.

They considered a game-theoretical model played between related a mother and female offspring (typically helper if sterile). Mother first decides whether she manipulates the offspring or not, and second, offspring decides whether she resists against the manipulation or not given that the mother expressed manipulation. With the decision for no resistance, offspring helps the mother. This is the structure of the game, and relatedness-weighted payoff is calculated (Fig 1C), as a naïve means to make the game-structure clearer. The condition for a given strategy to be favored

is generically written in Hamilton's form: $rB-C$ (r : genetic relatedness, B : fitness benefit an actor offered; C : fitness cost to the actor). I use this notation below.

The authors extended this model to the second scenario (Fig 1D). This scenario is clearly explained in the introduction (see the paragraph from L40):

- (1) the mother manipulates offspring to be helpers;
- (2) despite a possible resistance of the offspring against the manipulation, the mother can exploit the help provided from the helpers for a higher fecundity; and
- (3) it is the increased fecundity that allows them to reach agreement (in terms of $RB-C$, B increases so large that both individuals' traits are subject to the same direction of selection).

They then applies the "evolutionary" game theory (adaptive dynamics) by introducing a rare mutant (which has slightly a different trait value), which allows one to avoid using a naïve, relatedness-weighted payoffs, rather showing the Hamilton's rule in a more consistent way (eqs 1), from which the authors assessed the condition for the conflict dissolution.

The authors find that maternal manipulation of offspring helping allows the mother to increase her fertility beyond the threshold of parent-offspring conflict thereby reaching parent-offspring agreement.

Although this take-home message is very simple, I found a number of issues. First, the presentation of the result is very complicated. This complication starts from Figure 2, which appears to be elaborate and overcomplicated (containing too much information). I describe my concerns point-by-point, and I hope the editor and authors find my comments useful.

General:

Figs (especially fig 2): I found the figures very elaborated, but I am not sure if they really are clear. Simply, there is too much information packed into each single figure. As for fig 2, for instance, I spent a difficult time grasping the meaning of the entire figure. The meaning of f_2^{*} (dashed green line) is not explained enough, and the two, opposing arrows (pink) are really not needed. The y-axis is labelled as later fertility f_2 , but I did not understand how to read this figure. Is this a phase portrait? I do not have suggestion for an alternative though, the figure has much room for simplification. This issues may apply to other figures with heavy information-loads. More explanation for how to read the figures is needed.

The biological support or implication for the predictions/hypothesis is definitely needed. I am not saying that evidence is required, but that at least in-depth assessment of biological systems (e.g., hymenoptera) is needed; the discussion is currently superficial and the reader may wonder how one can test or confirm the predictions, how general this prediction is, and if this prediction has good support. Much room is available for discussion.

Genetic variation (GV): I agree that GV is important for the speed of adaptation. But what makes the difference in GV per se, in the system you're interested in? Why is it supposedly pronounced in hymenoptera? Why not in other systems?

Alleviation condition (Line 121-): This section is heavily theoretical, with few biological implications, although the mathematical meaning is well explained, which is not what readers want to read. Honestly, the epsilon (elasticity; eqn2) didn't help, as evolutionary models often exhibit this elasticity (although not fully mentioned in the literature).

Finally, Taylor-Frank approach (Line 273): The authors turn their attention to the theoretical limitations of Taylor-Frank (TF) approach (Taylor & Frank 1996, JTB). I tend to disagree with the description here. To my understanding, the modern TF approach encompasses broader classes of analytical approach than the original formulation (Rousset & Billiard 2000JEB, Ajar 2003, Rousset

2004 book, Taylor et al 2007 JEB, Gardner West & Wild 2011 JEB, etc. Mullon, Ohtsuki, and Lehmann's recent studies have further expanded the methodology). The starting point of the TF is indeed "recipient-based", that is, we count the fitness of a focal individual who is a recipient; however, this approach turns out to be equivalent to the "actor-based" approach (Taylor et al 2007 JEB), contrastingly to the authors' statement. Good examples for the TF methodology are found in Johnstone Cant & Field 2012 ProcB, and Lehmann Ravigne & Keller 2008 ProcB (in which they examined the inclusive fitness effects of producing sterile workers by mothers, and the inclusive fitness effects of being sterile workers; by the way these papers are relevant and I was surprised these were not even cited). More generally, the modern TF approach allows one to consider an allele that can be "silent" (e.g. when residing in males) but is active (e.g. when residing in females) in class-structured populations. This calculation is indeed tricky and technical, but TF approach can take recipient- and actor-based approaches both in a consistent way (see Taylor et al 2007 JEB). I think Mullon et al AmNat2016, NEE2018, etc provide good references for the authors. The present paper is not technically strict or rigorous and therefore does not offer mathematical proof of modeling kin selection effects. Finally, "Who helps whom" issue is interesting though, computing the kappa coefficient (Lehmann-Rousset 2010, PhilTransB) may help identify when natural selection favours which classes of individuals engage in helping whom. So honestly, I didn't get any points from this entire paragraph. I would remove this entire paragraph.

Minor:

Abstract "This mechanism is widely applicable", explain why? I was not convinced even from the main text (cf. Line 160). The wide applicability is not explained anywhere (but is merely stated a couple of times).

Hamilton's rule (L103): The authors referred to as "marginal" costs and benefits around here, but are you sure? I would say instead that the logarithmic derivatives would be the appropriate expressions for the marginal effects of fitness (correct me if I'm wrong; Frank 1998 book, equation 2.11).

Fig1: Information heavily loaded. I would for instance remake the figures like "Q-A" type diagram: first Q is "influence?", to which Yes and No comes with the arrows. The second Q is "resist?", and so on. Of course up to the authors' preference though, schematic figures should make the meaning very clear (in this regard Panels A and B are both excellent).

L 21: Definitions for B and C are neither precise. Hamilton didn't the rule that way. State that these are fitness effects.

L112 ("This matches the standard expectation"): I think that with only the single article (21: Keller & Nonacs) cited here, the majority of readers may wonder if this really is the standard expectation.

L198: This explanation for resolution vs dissolution should appear earlier, like in the introduction.

Sincerely,

[Dated: 6th Jan 2021]

Decision letter (RSPB-2020-2980.R0)

21-Jan-2021

Dear Dr González-Forero:

I am writing to inform you that your manuscript RSPB-2020-2980 entitled "Eusociality through conflict dissolution" has, in its current form, been rejected for publication in Proceedings B.

This action has been taken on the advice of referees, who have recommended that substantial revisions are necessary. With this in mind we would be happy to consider a resubmission, provided the comments of the referees are fully addressed. However please note that this is not a provisional acceptance.

Sincerely,
Professor Hans Heesterbeek
<mailto:proceedingsb@royalsociety.org>

Reviewer(s)' Comments to Author:

Referee: 1

Comments to the Author(s)

This paper proposes and models a pathway for the evolution of eusociality. Most research in this area has focused on kin selection, with a minority emphasizing parental manipulation. Those are often treated as mutually exclusive explanation, but it also seems possible that some combination of them is required. This paper explores one such way of combining them. Here, the first step is parental manipulation to get offspring to stay and help against their kin-selected interests. In the next step, there are two options. In one, the offspring evolve to resist the manipulation and the incipient eusociality disappears. In the other, the mother (or mother-father pair) evolve towards optimizing her life history to the new situation of having some helpers. This, in some situations, alters the kin-selected B/C ratio of the offspring such that it now pays for them to stay and help.

Eusociality is then selectively favored with no conflict, though the parental manipulation may persist as an honest signal.

The advantages of this model are nicely summarized by the authors. The initial threshold for manipulated helping is easier to achieve than the offspring kin selected one. Offspring acquiescence (under the altered B/C ratio) is most likely to be favored with high relatedness, consistent with the finding that single mating is the rule at the origin of eusocial taxa. The transition to honest signals from parental manipulation mechanisms is consistent with the kinds of signals observed in current social insects (although of course they could alternatively be honest signals without a manipulation precursor). And finally, a great advantage over the straight manipulation model is that the incentive for offspring to reverse the manipulation disappears and everyone agrees that helping is good. Overall, this model provides a potentially significant advance on a long-standing important question.

I think related ideas are probably out there in the literature, but maybe the idea has just been in my own head, as I'm not sure what citations to reference. One such idea is that maternal manipulation may immediately shift offspring choices such that helping is now favored. For example, by feeding an offspring less, the offspring may now be more likely to fail as a reproductive and, provided she is not equally more likely to fail as a helper (Craig 1979, 1983), that can make it pay to help instead. Craig was addressing West-Eberhard's "subfertility" hypothesis and I don't remember if they were explicitly invoking manipulation or not, but those papers might lead to others that do. Nevertheless, I see the current paper as a bit different. Instead of manipulation kicking offspring directly into the helping zone, it requires some further evolution of life history in the model.

I do not have any major reservations about the paper. Here are some minor comments:

Abstract (and elsewhere). The meaning of "requires that helpers alleviate the total percent life-history trade-off limiting maternal fertility" is not very clear to me. Can this be reworded in some way?

Figure 1. Good figure in general, though I don't think K is defined.

consistent with single-mating origins in a way that total parental manipulation is not

I have some issues with the treatment of other models in the Discussion. However, none of these issues detracts from the main conclusions of this paper.

You argue that rigorous derivation is needed because inclusive fitness theory can mislead (line 259 ff). That is certainly true; using inclusive fitness thinking can be tricky. But I am not convinced by your example of the monogamy hypothesis. I think the differing conclusions of your model relative to the monogamy hypothesis represent differences in assumptions. In your two-generation cycle, the decision to reproduce versus help in generation 1 means that either you or some additional siblings will be reproductives in generation 2. Therefore the relevant r 's are 1 and $\frac{1}{2}$. But I think proponents of the monogamy hypothesis are envisioning a different life history. Suppose you emerge in your mother's nest and your choice is 1) leave and begin reproducing now or 2) stay and begin rearing your siblings now. The relevant exchange is offspring for siblings and the relevant r 's are $\frac{1}{2}$ and $\frac{1}{2}$. In other words, in your model, the option of reproducing directly involves a delay; you wait for the next generation (instead of rearing siblings in this generation). The upshot of that, with respect to relatedness, is that your relatedness through sibling help gets diluted by $\frac{1}{2}$ by the extra generation.

So I think the value of rigorous models versus inclusive fitness is more nuanced. Yes, in some sense the last word rests with the rigorous models, if they get their assumptions right and do not falsely exclude reasonable alternatives. But those models can be so complex (most kin selection researchers will not be able to follow your models in detail) that we run the risk of taking their

results as more definitive than they are. Inclusive fitness reasoning, because it is so simple (though, yes, sometimes tricky), can provide a check on the rigorous models as well as a way for non-mathematicians to understand their results.

Parenthetically, as you will probably realize, your conflict dissolution ideas should not be ruined by the life history assumed by the monogamy model. There is still conflict because the mother's r 's have to be adjusted too (r to reproducing offspring $\frac{1}{2}$; r to that offspring's offspring $\frac{1}{4}$).

With perhaps somewhat less confidence, I would suggest that similar arguments might apply to your conclusions about the Taylor-Frank method (line 273 ff). This is suggested by your statement that it is unable to identify the logic problem with the monogamy hypothesis, which I have argued is not a logic problem at all. Again, I suspect that is true that Taylor-Frank modelers "subjective" choices about fitness are not necessarily subjective but may reflect assumptions of their model, though perhaps not always explicitly stated. Again, I would agree that it is great to have more rigorous models, but that they too can be misleading. Also, you are right that the T-F approach takes a direct-fitness perspective, but doesn't yours also take such a perspective from the beginning? Don't all "rigorous" models start that way and then have to reason themselves into a shifted inclusive perspective. That is commonly done with the T-F method as well.

Referee: 2

Comments to the Author(s)

Dear Editor, Proceedings B,

The paper entitled "Eusociality through conflict dissolution" by Gonzalez-Foreo and Pena is a theoretical paper investigating the origin of eusociality (sterile individuals helping the reproductives produce offspring under the division of labor, with cooperative interactions and overlapping generations). One of the hypotheses for the evolution of eusociality posits that in an evolutionary timescale, non-reproductive individuals (hereafter referred to as "helpers") originally resulted from the reproductives ("queens") manipulating them but at some point switched to work voluntarily: the authors mentioned this as "the converted helping hypothesis." One of the prerequisites of this hypothesis is queen-helper (or more specifically parent-offspring) has reached an agreement over evolutionary conflicts. Because of relatedness asymmetry, the direction of selection for the voluntary working may differ for the two (or more) classes of individuals, but if the direction of selection is (always) the same between the classes, the conflict of interests would no longer exist. Their question is how this agreement can reach and if this mechanism may be the origin of eusociality.

They considered a game-theoretical model played between related a mother and female offspring (typically helper if sterile). Mother first decides whether she manipulates the offspring or not, and second, offspring decides whether she resists against the manipulation or not given that the mother expressed manipulation. With the decision for no resistance, offspring helps the mother. This is the structure of the game, and relatedness-weighted payoff is calculated (Fig 1C), as a naïve means to make the game-structure clearer. The condition for a given strategy to be favored is generically written in Hamilton's form: $rB > C$ (r : genetic relatedness, B : fitness benefit an actor offered; C : fitness cost to the actor). I use this notation below.

The authors extended this model to the second scenario (Fig 1D). This scenario is clearly explained in the introduction (see the paragraph from L40):

- (1) the mother manipulates offspring to be helpers;
- (2) despite a possible resistance of the offspring against the manipulation, the mother can exploit the help provided from the helpers for a higher fecundity; and
- (3) it is the increased fecundity that allows them to reach agreement (in terms of $rB > C$, B increases so large that both individuals' traits are subject to the same direction of selection).

They then applies the "evolutionary" game theory (adaptive dynamics) by introducing a rare mutant (which has slightly a different trait value), which allows one to avoid using a naïve,

relatedness-weighted payoffs, rather showing the Hamilton's rule in a more consistent way (eqs 1), from which the authors assessed the condition for the conflict dissolution.

The authors find that maternal manipulation of offspring helping allows the mother to increase her fertility beyond the threshold of parent-offspring conflict thereby reaching parent-offspring agreement.

Although this take-home message is very simple, I found a number of issues. First, the presentation of the result is very complicated. This complication starts from Figure 2, which appears to be elaborate and overcomplicated (containing too much information). I describe my concerns point-by-point, and I hope the editor and authors find my comments useful.

General:

Figs (especially fig 2): I found the figures very elaborated, but I am not sure if they really are clear. Simply, there is too much information packed into each single figure. As for fig 2, for instance, I spent a difficult time grasping the meaning of the entire figure. The meaning of f_2^* (dashed green line) is not explained enough, and the two, opposing arrows (pink) are really not needed. The y-axis is labelled as later fertility f_2 , but I did not understand how to read this figure. Is this a phase portrait? I do not have suggestion for an alternative though, the figure has much room for simplification. This issues may apply to other figures with heavy information-loads. More explanation for how to read the figures is needed.

The biological support or implication for the predictions/hypothesis is definitely needed. I am not saying that evidence is required, but that at least in-depth assessment of biological systems (e.g., hymenoptera) is needed; the discussion is currently superficial and the reader may wonder how one can test or confirm the predictions, how general this prediction is, and if this prediction has good support. Much room is available for discussion.

Genetic variation (GV): I agree that GV is important for the speed of adaptation. But what makes the difference in GV per se, in the system you're interested in? Why is it supposedly pronounced in hymenoptera? Why not in other systems?

Alleviation condition (Line 121-): This section is heavily theoretical, with few biological implications, although the mathematical meaning is well explained, which is not what readers want to read. Honestly, the epsilon (elasticity; eqn2) didn't help, as evolutionary models often exhibit this elasticity (although not fully mentioned in the literature).

Finally, Taylor-Frank approach (Line 273): The authors turn their attention to the theoretical limitations of Taylor-Frank (TF) approach (Taylor & Frank 1996, JTB). I tend to disagree with the description here. To my understanding, the modern TF approach encompasses broader classes of analytical approach than the original formulation (Rousset & Billiard 2000 JEB, Ajar 2003, Rousset 2004 book, Taylor et al 2007 JEB, Gardner West & Wild 2011 JEB, etc. Mullon, Ohtsuki, and Lehmann's recent studies have further expanded the methodology). The starting point of the TF is indeed "recipient-based", that is, we count the fitness of a focal individual who is a recipient; however, this approach turns out to be equivalent to the "actor-based" approach (Taylor et al 2007 JEB), contrastingly to the authors' statement. Good examples for the TF methodology are found in Johnstone Cant & Field 2012 ProcB, and Lehmann Ravigne & Keller 2008 ProcB (in which they examined the inclusive fitness effects of producing sterile workers by mothers, and the inclusive fitness effects of being sterile workers; by the way these papers are relevant and I was surprised these were not even cited). More generally, the modern TF approach allows one to consider an allele that can be "silent" (e.g. when residing in males) but is active (e.g. when residing in females) in class-structured populations. This calculation is indeed tricky and technical, but TF approach can take recipient- and actor-based approaches both in a consistent way (see Taylor et al 2007 JEB). I think Mullon et al AmNat2016, NEE2018, etc provide good references for the

authors. The present paper is not technically strict or rigorous and therefore does not offer mathematical proof of modeling kin selection effects. Finally, "Who helps whom" issue is interesting though, computing the kappa coefficient (Lehmann-Rousset 2010, PhilTransB) may help identify when natural selection favours which classes of individuals engage in helping whom. So honestly, I didn't get any points from this entire paragraph. I would remove this entire paragraph.

Minor:

Abstract "This mechanism is widely applicable", explain why? I was not convinced even from the main text (cf. Line 160). The wide applicability is not explained anywhere (but is merely stated a couple of times).

Hamilton's rule (L103): The authors referred to as "marginal" costs and benefits around here, but are you sure? I would say instead that the logarithmic derivatives would be the appropriate expressions for the marginal effects of fitness (correct me if I'm wrong; Frank 1998 book, equation 2.11).

Fig1: Information heavily loaded. I would for instance remake the figures like "Q-A" type diagram: first Q is "influence?", to which Yes and No comes with the arrows. The second Q is "resist?", and so on. Of course up to the authors' preference though, schematic figures should make the meaning very clear (in this regard Panels A and B are both excellent).

L 21: Definitions for B and C are neither precise. Hamilton didn't the rule that way. State that these are fitness effects.

L112 ("This matches the standard expectation"): I think that with only the single article (21: Keller & Nonacs) cited here, the majority of readers may wonder if this really is the standard expectation.

L198: This explanation for resolution vs dissolution should appear earlier, like in the introduction.

Sincerely,
[Dated: 6th Jan 2021]

Author's Response to Decision Letter for (RSPB-2020-2980.R0)

See Appendix A.

RSPB-2021-0386.R0

Review form: Reviewer 2

Recommendation

Accept with minor revision (please list in comments)

We now publish an annual list of reviewers in the journal. The list contains names and ORCID (when provided) but does not disclose which article you reviewed. Does your co-reviewer wish to be included in the list?

No

Scientific importance: Is the manuscript an original and important contribution to its field?

Good

General interest: Is the paper of sufficient general interest?

Good

Quality of the paper: Is the overall quality of the paper suitable?

Good

Is the length of the paper justified?

Yes

Should the paper be seen by a specialist statistical reviewer?

No

Do you have any concerns about statistical analyses in this paper? If so, please specify them explicitly in your report.

No

It is a condition of publication that authors make their supporting data, code and materials available - either as supplementary material or hosted in an external repository. Please rate, if applicable, the supporting data on the following criteria.

Is it accessible?

Yes

Is it clear?

Yes

Is it adequate?

Yes

Do you have any ethical concerns with this paper?

No

Comments to the Author

Dear Editor,

Please convey my apology to the authors for the delay in my report.

I read the revised manuscript and found it well revised and therefore a significant improvement over the previous.

Congratulations for the authors.

I have a couple of very minor issues. The line numbers are all from the "clean" version of their manuscript.

Line22 (ref9). Is this voluntary helping hypothesis supported or not?

Line165- Discussion. The first paragraph ends with a repetition of some of the introduction (Line35). Could be rewritten based on the predictions from your models.

Line174: "being supported by ... supporting both..." reads a bit weird. (A is supported by B which supports C). Could be rephrased.

Line185: destabilize the eusociality - true, but could be rewritten as "... that would then not lead to eusociality" or equivalent. Destabilization is a polysemous.

Line196: ESSs needs citation, Maynard Smith Price?

Figure4: I am not really able to tell yellow from green. Using more visible and universally designed colors would be appreciated....

Decision letter (RSPB-2021-0386.R0)

19-Mar-2021

Dear Dr González-Forero

I am pleased to inform you that your manuscript RSPB-2021-0386 entitled "Eusociality through conflict dissolution" has been accepted for publication in Proceedings B.

The referee and the Associate Editor have recommended publication, but also suggest some minor revisions to your manuscript. Therefore, I invite you to respond to the referee's comments and revise your manuscript. Because the schedule for publication is very tight, it is a condition of publication that you submit the revised version of your manuscript within 7 days. If you do not think you will be able to meet this date please let us know.

Sincerely,

Professor Hans Heesterbeek

Associate Editor

Board Member

Comments to Author:

Thank you for your revision. We have some minor comments from one of the referees that you should address. Overall I think the paper is an important contribution and I am pleased that you have taken out the discussion of the relative merits of different approaches. That conversation has somewhat overshadowed the field for a while. My sense is that this is an important contribution and will be widely influential.

Reviewer(s)' Comments to Author:

Referee: 2

Comments to the Author(s).

Dear Editor,

Please convey my apology to the authors for the delay in my report.

I read the revised manuscript and found it well revised and therefore a significant improvement over the previous.

Congratulations for the authors.

I have a couple of very minor issues. The line numbers are all from the "clean" version of their manuscript.

Line22 (ref9). Is this voluntary helping hypothesis supported or not?

Line165- Discussion. The first paragraph ends with a repetition of some of the introduction (Line35). Could be rewritten based on the predictions from your models.

Line174: "being supported by ... supporting both..." reads a bit weird. (A is supported by B which supports C). Could be rephrased.

Line185: destabilize the eusociality - true, but could be rewritten as "... that would then not lead to eusociality" or equivalent. Destabilization is a polysemy.

Line196: ESSs needs citation, Maynard Smith Price?

Figure4: I am not really able to tell yellow from green. Using more visible and universally designed colors would be appreciated....

Author's Response to Decision Letter for (RSPB-2021-0386.R0)

See Appendix B.

Decision letter (RSPB-2021-0386.R1)

23-Mar-2021

Dear Dr González-Forero

I am pleased to inform you that your manuscript entitled "Eusociality through conflict dissolution" has been accepted for publication in Proceedings B.

Data Accessibility section

Open Access

Paper charges

Sincerely,

Proceedings B

Appendix A

Response to Editor and Reviewers' comments

Reviewer 1

This paper proposes and models a pathway for the evolution of eusociality. Most research in this area has focused on kin selection, with a minority emphasizing parental manipulation. Those are often treated as mutually exclusive explanation, but it also seems possible that some combination of them is required. This paper explores one such way of combining them. Here, the first step is parental manipulation to get offspring to stay and help against their kin-selected interests. In the next step, there are two options. In one, the offspring evolve to resist the manipulation and the incipient eusociality disappears. In the other, the mother (or mother-father pair) evolve towards optimizing her life history to the new situation of having some helpers. This, in some situations, alters the kin-selected B/C ratio of the offspring such that it now pays for them to stay and help. Eusociality is then selectively favored with no conflict, though the parental manipulation may persist as an honest signal.

The advantages of this model are nicely summarized by the authors. The initial threshold for manipulated helping is easier to achieve than the offspring kin selected one. Offspring acquiescence (under the altered B/C ratio) is most likely to be favored with high relatedness, consistent with the finding that single mating is the rule at the origin of eusocial taxa. The transition to honest signals from parental manipulation mechanisms is consistent with the kinds of signals observed in current social insects (although of course they could alternatively be honest signals without a manipulation precursor). And finally, a great advantage over the straight manipulation model is that the incentive for offspring to reverse the manipulation disappears and everyone agrees that helping is good. Overall, this model provides a potentially significant advance on a long-standing important question.

We are glad that the reviewer enjoyed the paper and that they think our model provides an advance on an important question. We are also very grateful for their suggestions.

I think related ideas are probably out there in the literature, but maybe the idea has just been in my own head, as I'm not sure what citations to reference. One such idea is that maternal manipulation may immediately shift offspring choices such that helping is now favored. For example, by feeding an offspring less, the offspring may now be more likely to fail as a reproductive and, provided she is not equally more likely to fail as a helper (Craig 1979, 1983), that can make it pay to help instead. Craig was addressing West-Eberhard's "subfertility" hypothesis and I don't remember if they were explicitly invoking manipulation or not, but those papers might lead to others that do. Nevertheless, I see the current paper as a bit different. Instead of manipulation kicking offspring directly into the helping zone, it requires some further evolution of life history in the model.

The idea that the mother feeds offspring less is indeed relevant, and one of us has previously addressed it. Specifically, when manipulation is executed in a way that it is costly to resist (e.g., by feeding offspring poorly, as suggested by the reviewer, West-Eberhard 1975 and Craig 1979, and originally by Alexander 1974), González-Forero & Gavrillets (2013; eq. 6b) showed that acquiescence to manipulation is favored if $br > c_a - c_r$, which is a Hamilton's rule where the cost of helping is the difference between the cost of acquiescence c_a and the cost of resistance c_r (similar equations have been found in other contexts; e.g., Pagel et al. 1998. Reduced parasitism by retaliatory cuckoos selects for hosts that rear cuckoo nestlings. *Behavioral Ecology* 9:566–572.). So, as the reviewer suggests, manipulation can immediately alter offspring choices if resistance is costly. However, we are here interested in the situation where manipulation is executed in a more subtle way (i.e., queen pheromones), which has long been seen as rendering manipulation unable to yield stable eusociality (e.g., see Craig 1979 p. 332 or Keller and Nonacs 1993). Such view implicitly assumes that resistance is rather costless, and our interest is to understand if under such unfavorable conditions for manipulation, it can still yield stable eusociality. One of us previously tackled this situation and found that manipulation can indeed yield stable eusociality if resistance is costless, but the mechanism studied required conditions that are not immediately supported by available empirical evidence (i.e., that offspring that do not receive maternal care are better users of help than those that do receive maternal care; González-Forero 2015). Consequently, as the reviewer suggests, our paper is indeed different in that we here present a mechanism that enables the origin of stable eusociality from manipulation, where resistance to manipulation can be costless (e.g., if queen pheromones can be ignored at no cost), and where the required conditions

(i.e., that helpers alleviate trade-offs faced by their mother) are of wide relevance as indicated by available evidence (we explain this empirical support better in this revision in L 226-232 of highlighted revision file, motivated by the comments of reviewer 2).

I do not have any major reservations about the paper. Here are some minor comments:

Abstract (and elsewhere). The meaning of “requires that helpers alleviate the total percent life-history trade-off limiting maternal fertility” is not very clear to me. Can this be reworded in some way?

We reworded this in various places. In particular, in the abstract we now use the simpler sentence “requires that helpers alleviate maternal life-history trade-offs” and reworded analogous expressions in the main text, also following comments from reviewer 2. In some places, we have kept the phrase quoted by the reviewer because it is the technically most precise.

Figure 1. Good figure in general, though I don't think K is defined.

Thanks for pointing this out. We previously defined K in the text (L 68 of highlighted revision file), but now also define it at its earliest occurrence at the end of the figure 1 legend.

consistent with single-mating origins in a way that total parental manipulation is not

I have some issues with the treatment of other models in the Discussion. However, none of these issues detracts from the main conclusions of this paper.

You argue that rigorous derivation is needed because inclusive fitness theory can mislead (line 259 ff). That is certainly true; using inclusive fitness thinking can be tricky. But I am not convinced by your example of the monogamy hypothesis. I think the differing conclusions of your model relative to the monogamy hypothesis represent differences in assumptions. In your two-generation cycle, the decision to reproduce versus help in generation 1 means that either you or some additional siblings will be reproductives in generation 2. Therefore the relevant r 's are 1 and $\frac{1}{2}$. But I think proponents of the monogamy hypothesis are envisioning a different life history. Suppose you emerge in your mother's nest and your choice is 1) leave and begin reproducing now or 2) stay and begin rearing your siblings now. The relevant exchange is offspring for siblings and the relevant r 's are $\frac{1}{2}$ and $\frac{1}{2}$. In other words, in your model, the option of reproducing directly involves a delay; you wait for the next generation (instead of rearing siblings in this generation). The upshot of that, with respect to relatedness, is that your relatedness through sibling help gets diluted by $\frac{1}{2}$ by the extra generation.

We thank the reviewer for this insight. We agree that it is unnecessary to dwell on this point for the interests of the paper, and have opted for removing the paragraph discussing the monogamy hypothesis.

So I think the value of rigorous models versus inclusive fitness is more nuanced. Yes, in some sense the last word rests with the rigorous models, if they get their assumptions right and do not falsely exclude reasonable alternatives. But those models can be so complex (most kin selection researchers will not be able to follow your models in detail) that we run the risk of taking their results as more definitive than they are. Inclusive fitness reasoning, because it is so simple (though, yes, sometimes tricky), can provide a check on the rigorous models as well as a way for non-mathematicians to understand their results.

Parenthetically, as you will probably realize, your conflict dissolution ideas should not be ruined by the life history assumed by the monogamy model. There is still conflict because the mother's r 's have to be adjusted too (r to reproducing offspring $\frac{1}{2}$; r to that offspring's offspring $\frac{1}{4}$).

With perhaps somewhat less confidence, I would suggest that similar arguments might apply to your conclusions about the Taylor-Frank method (line 273 ff). This is suggested by your statement that it is unable to identify the logic problem with the monogamy hypothesis, which I have argued is not a logic problem at all. Again, I suspect that is true that Taylor-Frank modelers "subjective" choices about fitness are not necessarily subjective but may reflect assumptions of their model, though perhaps not always explicitly stated. Again, I would agree that it is great to have more rigorous models, but that they too can be misleading. Also, you are right that the T-F approach takes a direct-fitness perspective,

but doesn't yours also take such a perspective from the beginning? Don't all "rigorous" models start that way and then have to reason themselves into a shifted inclusive perspective. That is commonly done with the T-F method as well.

Thanks for your comments on this. We see that the points we made regarding Taylor and Frank were interpreted by both reviewer 1 and 2 differently from what we intended. To avoid the risk of finding ourselves trapped in a semantic debate, we have opted for removing the paragraph discussing how our approach differs from Taylor and Frank, as suggested by reviewer 2. This addresses the concerns of both reviewers.

Reviewer 2

The paper entitled “Eusociality through conflict dissolution” by Gonzalez-Foreo and Pena is a theoretical paper investigating the origin of eusociality (sterile individuals helping the reproductives produce offspring under the division of labor, with cooperative interactions and overlapping generations). One of the hypotheses for the evolution of eusociality posits that in an evolutionary timescale, non-reproductive individuals (hereafter referred to as “helpers”) originally resulted from the reproductives (“queens”) manipulating them but at some point switched to work voluntarily: the authors mentioned this as “the converted helping hypothesis.” One of the prerequisites of this hypothesis is queen-helper (or more specifically parent-offspring) has reached an agreement over evolutionary conflicts. Because of relatedness asymmetry, the direction of selection for the voluntary working may differ for the two (or more) classes of individuals, but if the direction of selection is (always) the same between the classes, the conflict of interests would no longer exist. Their question is how this agreement can reach and if this mechanism may be the origin of eusociality.

They considered a game-theoretical model played between related mother and female offspring (typically helper if sterile). Mother first decides whether she manipulates the offspring or not, and second, offspring decides whether she resists against the manipulation or not given that the mother expressed manipulation. With the decision for no resistance, offspring helps the mother. This is the structure of the game, and relatedness-weighted payoff is calculated (Fig 1C), as a naïve means to make the game-structure clearer. The condition for a given strategy to be favored is generically written in Hamilton’s form: $rB > C$ (r : genetic relatedness, B : fitness benefit an actor offered; C : fitness cost to the actor). I use this notation below.

The authors extended this model to the second scenario (Fig 1D). This scenario is clearly explained in the introduction (see the paragraph from L40): (1) the mother manipulates offspring to be helpers; (2) despite a possible resistance of the offspring against the manipulation, the mother can exploit the help provided from the helpers for a higher fecundity; and (3) it is the increased fecundity that allows them to reach agreement (in terms of $rB > C$, B increases so large that both individuals’ traits are subject to the same direction of selection).

They then apply the “evolutionary” game theory (adaptive dynamics) by introducing a rare mutant (which has slightly a different trait value), which allows one to avoid using a naïve, relatedness-weighted payoffs, rather showing the Hamilton’s rule in a more consistent way (eqs 1), from which the authors assessed the condition for the conflict dissolution.

The authors find that maternal manipulation of offspring helping allows the mother to increase her fertility beyond the threshold of parent-offspring conflict thereby reaching parent-offspring agreement.

Although this take-home message is very simple, I found a number of issues. First, the presentation of the result is very complicated. This complication starts from Figure 2, which appears to be elaborate and overcomplicated (containing too much information). I describe my concerns point-by-point, and I hope the editor and authors find my comments useful.

We are very grateful for the reviewer’s evaluation of and comments on our manuscript. We have sought to incorporate their comments as much as possible.

General:

Figs (especially fig 2): I found the figures very elaborated, but I am not sure if they really are clear. Simply, there is too much information packed into each single figure. As for fig 2, for instance, I spent a difficult time grasping the meaning of the entire figure. The meaning of f_2^* (dashed green line) is not explained enough, and the two, opposing arrows (pink) are really not needed. The y-axis is labelled as later fertility f_2 , but I did not understand how to read this figure. Is this a phase portrait? I do not have suggestion for an alternative though, the figure has much room for simplification. This issues may apply to other figures with heavy information-loads. More explanation for how to read the figures is needed.

The reviewer is right that further explanation of how to read Fig. 2 was needed and that f_2^* was previously defined rather late (L 145 in highlighted revision file). We have added explanations in the figure legend, including

the meaning of f_2^* at its earliest occurrence, how to read the figures, and whether they are phase portraits. We also relocated the two opposing arrows to make them clearer. We believe that Fig. 2 is now easier to understand.

The biological support or implication for the predictions/hypothesis is definitely needed. I am not saying that evidence is required, but that at least in-depth assessment of biological systems (e.g., hymenoptera) is needed; the discussion is currently superficial and the reader may wonder how one can test or confirm the predictions, how general this prediction is, and if this prediction has good support. Much room is available for discussion.

Thank you for pointing this out. We have now added a paragraph to discuss empirical support for the hypothesis (L 226-232 in highlighted revision file). This led us to change our wording from stating that our mechanism was “widely applicable” to stating that its key requirement is seemingly supported by available evidence. This has been very helpful to clarify the relevance of this hypothesis.

Genetic variation (GV): I agree that GV is important for the speed of adaptation. But what makes the difference in GV per se, in the system you're interested in? Why is it supposedly pronounced in hymenoptera? Why not in other systems?

In this paper, we have not sought to suggest that genetic variation is relatively favorable for conflict dissolution in hymenoptera. We also have not aimed to address the question of why eusociality should be more prevalent in hymenoptera. In this paper, we have limited ourselves to show that conflict dissolution via maternal reproductive specialization can occur under diploidy and haplodiploidy. We think that that question of whether haplodiploidy promotes or inhibits conflict dissolution deserves a separate paper and thus have deliberately chose not to tackle it here.

Alleviation condition (Line 121-): This section is heavily theoretical, with few biological implications, although the mathematical meaning is well explained, which is not what readers want to read. Honestly, the epsilon (elasticity; eqn2) didn't help, as evolutionary models often exhibit this elasticity (although not fully mentioned in the literature).

We thank the reviewer for this comment. We have expanded the verbal descriptions in this section (L 142 and later, in the highlighted revision file). This has helped to make the biological meaning clearer. However, we appreciate that this section remains relatively technical. We have opted to keep some technicality in this section as its points are biologically important but can be misinterpreted if technicality is avoided. Specifically, this section gives the precise meaning of the trade-off alleviation requirement. We appreciate that many biologists may shy away from these mathematical expressions, and for them we have added verbal descriptions. Yet, we have kept some technicality for other readers that will be able to interpret these expressions and extract the precise meaning.

Finally, Taylor-Frank approach (Line 273): The authors turn their attention to the theoretical limitations of Taylor-Frank (TF) approach (Taylor & Frank 1996, JTB). I tend to disagree with the description here. To my understanding, the modern TF approach encompasses broader classes of analytical approach than the original formulation (Rousset & Billiard 2000 JEB, Ajar 2003, Rousset 2004 book, Taylor et al 2007 JEB, Gardner West & Wild 2011 JEB, etc. Mullan, Ohtsuki, and Lehmann's recent studies have further expanded the methodology). The starting point of the TF is indeed “recipient-based”, that is, we count the fitness of a focal individual who is a recipient; however, this approach turns out to be equivalent to the “actor-based” approach (Taylor et al 2007 JEB), contrastingly to the authors' statement. Good examples for the TF methodology are found in Johnstone Cant & Field 2012 ProcB, and Lehmann Ravigne & Keller 2008 ProcB (in which they examined the inclusive fitness effects of producing sterile workers by mothers, and the inclusive fitness effects of being sterile workers; by the way these papers are relevant and I was surprised these were not even cited). More generally, the modern TF approach allows one to consider an allele that can be “silent” (e.g. when residing in males) but is active (e.g. when residing in females) in class-structured populations. This calculation is indeed tricky and technical, but TF approach can take recipient- and actor-based approaches both in a consistent way (see Taylor et al 2007 JEB). I think Mullan et al AmNat2016, NEE2018, etc provide good references for the authors. The present paper is not technically strict or rigorous and therefore does not offer

mathematical proof of modeling kin selection effects. Finally, “Who helps whom” issue is interesting though, computing the kappa coefficient (Lehmann-Rousset 2010, PhilTransB) may help identify when natural selection favours which classes of individuals engage in helping whom. So honestly, I didn't get any points from this entire paragraph. I would remove this entire paragraph.

The reviewer interprets our comments about the Taylor and Frank approach (TF) as referring to the neighbor-modulated approach particularly in its formalized forms under adaptive dynamics frameworks (which we are well aware of, yet have underlying demographic models different to ours so we do not use their results here). Our comments referred to the original TF rather than later formalizations; the reviewer may note that the original TF rather than later formalizations continues to be widely used. We fear that our discussion of TF can be misinterpreted as indicated by the interpretation the reviewer perceived, so we have opted to remove that paragraph as suggested by the reviewer.

Minor:

Abstract “This mechanism is widely applicable”, explain why? I was not convinced even from the main text (cf. Line 160). The wide applicability is not explained anywhere (but is merely stated a couple of times).

We thank the reviewer for this comment. Motivated by this, we have added a discussion on empirical evidence for trade-off alleviation (L 226-232 in the highlighted revision file). This led to removing the phrase “widely applicable” to stating that available evidence across social hymenoptera and cooperatively breeding mammals and birds indicates that the key requirement of trade-off alleviation may hold widely across eusocial taxa.

Hamilton's rule (L103): The authors referred to as “marginal” costs and benefits around here, but are you sure? I would say instead that the logarithmic derivatives would be the appropriate expressions for the marginal effects of fitness (correct me if I'm wrong; Frank 1998 book, equation 2.11).

The quantities C and B are not logarithmic derivatives (they are not divided by early Π_1 or late Π_2 productivities, respectively). They are derivatives of early and late productivities with respect to helper number, that is, they are marginal costs and benefits.

Fig1: Information heavily loaded. I would for instance remake the figures like "Q-A" type diagram: first Q is "influence?", to which Yes and No comes with the arrows. The second Q is "resist?", and so on. Of course up to the authors' preference though, schematic figures should make the meaning very clear (in this regard Panels A and B are both excellent).

Although we thank the reviewer for this suggestion, Fig. 1B,C somewhat follows a standard approach to draw an extensive-form game, which we opt for preserving.

L 21: Definitions for B and C are neither precise. Hamilton didn't the rule that way. State that these are fitness effects.

As this is an introductory sentence, we think that complete technical precision at this point is neither desirable nor necessary.

L112 ("This matches the standard expectation"): I think that with only the single article (21: Keller & Nonacs) cited here, the majority of readers may wonder if this really is the standard expectation.

We added references to two more recent reviews (L 123 in highlighted revision file).

L198: This explanation for resolution vs dissolution should appear earlier, like in the introduction.

Thanks for this suggestion. We tried to do as indicated, but felt that the distinction between resolution and dissolution was a bit too technical for the introduction, and that deviated attention too much from the basic points we wanted to convey there. So we chose to keep the resolution vs dissolution paragraph in the discussion, although we moved it slightly earlier (L 213 in highlighted revision file).

Appendix B

Response to Editor and Reviewers' comments

Reviewer 2

Line22 (ref9). Is this voluntary helping hypothesis supported or not?

We are very grateful for the reviewer's evaluation of and comments on our manuscript. We added a few words at the end of the paragraph, so the last sentence of the paragraph now reads: "Although, by definition, the maternal manipulation hypothesis would account for the prevalence of maternal influence, this hypothesis is refuted by increasing evidence suggesting that it is often in the evolutionary interests of offspring to help, thus supporting the voluntary helping hypothesis".

Line165- Discussion. The first paragraph ends with a repetition of some of the introduction (Line35). Could be rewritten based on the predictions from your models.

We opted to keep this repetition as it serves to introduce the two following paragraphs in the discussion.

Line174: "being supported by ... supporting both..." reads a bit weird. (A is supported by B which supports C). Could be rephrased.

Unfortunately, we didn't find a better alternative phrasing.

Line185: destabilize the eusociality - true, but could be rewritten as "... that would then not lead to eusociality" or equivalent. Destabilization is a polysemous.

Thanks for this. We rephrased this sentence to end "that would prevent the evolution of eusociality".

Line196: ESSs needs citation, Maynard Smith Price?

We appreciate this suggestion, but we chose to leave this uncited because 1) we do not use the ESS notion in our analyses, 2) we have had to limit our references to a subset of works due to space constraints, and 3) the ESS notion can by now be relatively safely regarded as common knowledge in evolutionary biology.

Figure4: I am not really able to tell yellow from green. Using more visible and universally designed colors would be appreciated...

We thank the reviewer for this. Although we have already sought to use color-blind friendly colors throughout, we tried to heed the reviewer's indication by implementing a color-blind palette that is available in python which we use to produce our figures. However, this palette greatly reduces the intelligibility of other colors and we opted for not using it. We tested our figure 2 with the same colors we had in <https://mariechatfield.com/simple-pdf-viewer/> and found that the dashed green line can be distinguished under all color vision deficiencies listed. Consequently, we have opted to keep the same colors we had hoping this is not going to cause problems since the green color the reviewer refers to is only used in dashed lines in panels where no other dashed lines occur so dashing would hopefully be sufficient to distinguish the dashed lines from the background.